# A Multi-Modal Multilingual Benchmark for Document Image Classification

**Yoshinari Fujinuma**[*]  **Siddharth Varia**[*]  **Nishant Sankaran**
**Bonan Min**  **Srikar Appalaraju**  **Yogarshi Vyas**
AWS AI Labs
{fujinuy,siddhvar,nishank,bonanmin,srikara,yogarshi}@amazon.com

## Abstract

Document image classification is different from plain-text document classification and consists of classifying a document by understanding the content and structure of documents such as forms, emails, and other such documents. We show that the only existing dataset for this task (Lewis et al., 2006) has several limitations and we introduce two newly curated multilingual datasets (WIKI-DOC and MULTIEURLEX-DOC) that overcome these limitations. We further undertake a comprehensive study of popular visually-rich document understanding or Document AI models in previously untested setting in document image classification such as 1) multi-label classification, and 2) zero-shot cross-lingual transfer setup. Experimental results show limitations of multilingual Document AI models on cross-lingual transfer across typologically distant languages. Our datasets and findings open the door for future research into improving Document AI models.[1]

## 1 Introduction

Visual document understanding aims to extract useful information from a variety of documents (e.g., forms, tables) beyond merely Optical Character Recognition (OCR). Unlike texts in traditional NLP tasks, this task is challenging since the content is laid out in a 2D structure where each word contains an $(x, y)$ location in the document. Prior work has shown that this task requires a model to process information occurring in multiple modalities including cues in images and text as well as spatial cues (Xu et al., 2020). Visual document understanding consists of several sub-tasks including, but not limited to, document image classification (Lewis et al., 2006), entity extraction (Guillaume Jaume, 2019) and labeling (Park et al., 2019), and visual question answering (Mathew et al., 2020). Models for document understanding tasks have also benefited from large-scale unsupervised pretraining that infuses data from the various modalities (Appalaraju et al., 2021; Huang et al., 2022, *inter alia*).

In this work, we focus on the task of document classification on visually-rich documents, which aims to classify a given input document (usually PDF or an image) into one or more classes. A popular benchmark dataset for this task is the RVL-CDIP collection (Lewis et al., 2006) which consists of 16 different classes. However, this dataset is not designed to evaluate models on the document classification task with deeper understanding (Larson et al., 2023)—it consists of documents in English only, the documents are relevant to a single domain, each document belongs to only one class (i.e., multi-class), and some class labels (e.g., "email", "resume") do not require rich semantic understanding of the contents of the document.

Given these limitations, we argue that it is paramount to expand the evaluation for document classification to gain further insights into existing approaches, as well as identify their limitations. We propose two new datasets, MULTIEURLEX-DOC and WIKI-DOC, that complement RVL-CDIP in different ways. MULTIEURLEX-DOC consists of EU laws in 23 different languages and is a multi-label dataset where each document is assigned to one or more of different labels. Additionally, WIKI-DOC is a multi-class dataset comprises of rendered Wikipedia articles that covers non-European languages. These datasets are derived from prior work by Chalkidis et al. (2021) and Sinha et al. (2018) respectively. Both datasets require a deeper understanding of the text and contents of the documents to arrive at the correct label(s).

We use these new datasets to study the behavior of pre-trained visually-rich document understanding (or Document AI) models with a focus on answering three specific questions in document image

---

[*]Equal contribution.
[1]Code and dataset linked at https://huggingface.co/datasets/AmazonScience/MultilingualMultiModalClassification

| Dataset | Domain | Multilingual | Multi-Class | Multi-Label | Layout | Class Type | #Classes |
|---|---|---|---|---|---|---|---|
| RVL-CDIP | Tobacco Ind. | | ✓ | | Diverse | Doc. Type | 16 |
| MULTIEURLEX-DOC | EU Law | ✓ | | ✓ | Static | Content | 567 |
| WIKI-DOC | Wikipedia | ✓ | ✓ | | Static | Content | 111 |

Table 1: A comparison of the datasets introduced in this work with RVL-CDIP, the most commonly used dataset for document image classification. The newly curated datasets complement RVL-CDIP since they are multilingual, multi-label, and have classes based on document contents rather than document types (Doc. Type).

classification. First, focusing only on the English portions of the datasets, we examine if different pre-trained models perform consistently across datasets. Second, we ask whether multi-lingual document understanding models exhibit similar performances across different languages. Finally, we focus on the cross-lingual generalization ability of these models, and ask whether multi-lingual document understanding models can be used to classify documents without having to be trained on that specific language. We use a variety of pre-trained models that consume different inputs — text-only (Chi et al., 2021), text+layout (Xu et al., 2020, 2021b; Wang et al., 2022), multi-modal (Appalaraju et al., 2021; Xu et al., 2021a, 2022; Huang et al., 2022) that fuse input from text, layout, and visual features, as well as image-only models (Kim et al., 2022). Empirical results on the two new datasets reveal that image-only models have large improvement opportunities unlike what is reported on RVL-CDIP (Kim et al., 2022), and multi-modal models have limited cross-lingual generalization ability. Our main contributions are as follows:

- We introduce two new document image classification datasets which complement the domains and languages covered in the commonly used document image classification dataset (Lewis et al., 2006).
- We evaluate Document AI models on the newly created datasets and address limitations of such models on both multi-label and multi-class document classification tasks where understanding of document contents is crucial.
- We evaluate both the multilingual and cross-lingual generalization ability of Document AI models showing challenges in transferring across syntactically distant languages.

## 2 Related Work

Models designed for document image understanding tasks, often referred to as Document AI models, appear in various different forms. We give a brief overview in this section and leave the details to the survey paper by Cui (2021). Technically, any pre-trained text models (e.g., BERT or GPT-3) can be applied to handle document images after being processed by OCR tools. However, initial work by Xu et al. (2020) opened up the exploration of including additional modalities such as layouts and images for handling documents. Since then, many layout-aware models and pretraining tasks have been proposed in the literature (Li et al., 2021; Hong et al., 2022; Wang et al., 2022; Li et al., 2022a; Hao et al., 2022), followed by using images as additional input for better models that exploit a broader set of signals from the input (Appalaraju et al., 2021; Xu et al., 2021a; Huang et al., 2022; Biten et al., 2021; Appalaraju et al., 2023).

Recently, Document AI models using only images as inputs (or OCR-free models) have been attracting attention from researchers and practitioners (Rust et al., 2022; Kim et al., 2022; Lee et al., 2022). As a results of not being dependent on OCR, such models potentially avoid the propagation of OCR errors. Furthermore, these models do not have a fixed vocabulary and can technically be applied to any language without out-of-vocabulary concerns. Nevertheless, most models support only English, and only limited number of multilingual models have been explored in the literature. Additionally, such models are not thoroughly benchmarked on multilingual datasets due to lack of appropriate datasets. We address this by curating new multilingual datasets for document classification.

## 3 Multilingual Evaluation Datasets for Document AI Models

A common benchmark dataset for evaluating Document AI models on document classification tasks is RVL-CDIP (Lewis et al., 2006). However, we argue that using a single benchmark dataset naturally inhibits our understanding of the task as well as the solutions built for it. Specifically, we argue that RVL-CDIP has the following limitations:

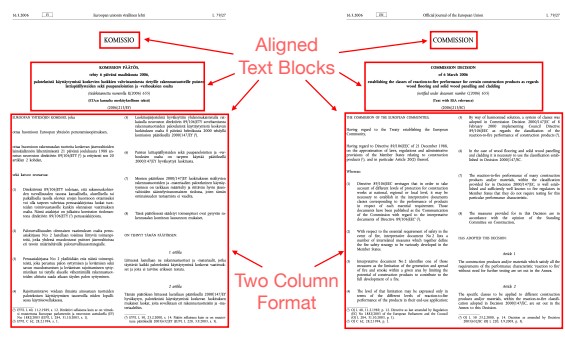
(a) MULTIEURLEX-DOC Examples

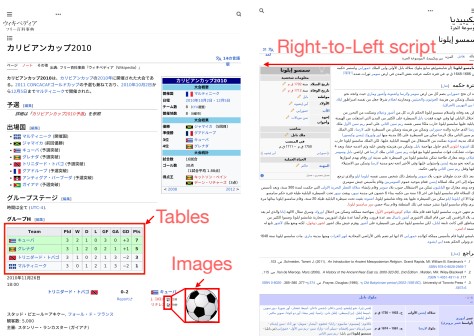
(b) WIKI-DOC Examples

Figure 1: Example documents from the newly curated datasets. MULTIEURLEX-DOC contains documents in different languages with highly-aligned layouts across European languages in multi-column format. WIKI-DOC contains documents with rich non-textual contents (e.g., tables and images) and documents in a broader set of languages, including Arabic which follows right-to-left writing.

1. It only includes English documents, thus limiting our understanding of multilingual document understanding models (Xu et al., 2021b) and their cross-lingual generalization ability.

2. The labels assigned to documents focus on the *type* of the document (e.g., "emails", "invoices", "tax form"), which does not necessarily require a deeper understanding of the contents of the documents.

Larson et al. (2023) provide an in-depth analysis on these limitations of RVL-CDIP and recommend that new datasets for document image classification should be multi-label to handle the naturally occurring overlap across labels, large-scale with 100+ classes, and multilingual to test the ability of models to transfer across languages (besides being accurately labeled). We introduce newly curated datasets covering these desired characteristics to complement the limitations of RVL-CDIP.

### 3.1 Newly Curated Datasets

We now introduce our two newly curated datasets and the summarized comparison between RVL-CDIP and the two newly curated datasets are in Table 1 and we give a summarized overview of the datasets in Figure 1.

**Multi-EurLex PDFs (MULTIEURLEX-DOC)**
Our first dataset is MULTIEURLEX-DOC, a multi-lingual and multi-label dataset consisting of European Union laws covering 23 European languages in their original PDF format. Documents in MULTIEURLEX-DOC consist of PDFs with layouts such as single column or multiple columns, and include many structural elements essential to

understanding the document such as headers, footers, and tables. Further, since the same law exists in multiple languages, layouts are also aligned across languages. Figure 1a shows an example of documents in this dataset. With this dataset, we aim to marry the recent progress in legal NLP (Hendrycks et al., 2021; Kementchedjhieva and Chalkidis, 2023; Chalkidis et al., 2022, *inter alia*) with the progress made in visually-rich document understanding.

Labels in MULTIEURLEX-DOC are derived from the EUROVOC taxonomy and are hierarchically organized into three increasingly specific levels.[2] For example, for the label 'Agri-foodstuffs' at level 1, the corresponding labels at level 2 are 'Plant Product' and 'Animal Product' where 'Plant Product' is further divided into 'Fruit', 'Vegetable' and 'Cereals' at level 3. We focus on evaluating models at level-3 (total of 567 classes) to make our results comparable to those reported in Chalkidis et al. (2021).

We put together MULTIEURLEX-DOC following a multi-step process: 1) We download the PDFs for all languages from EurLex website by using the CELEX ID[3] for each PDF obtained from dataset[4] released by Chalkidis et al. (2021) where it does not include PDFs. 2) We convert each page of the PDF into an image (currently, we only use image of first page of each document for modelling). 3) We apply OCR to extract words and bounding boxes for each image from the previous step. 4) Finally,

---

[2] http://eurovoc.europa.eu/

[3] The English example in Figure 1a is at https://eur-lex.europa.eu/legal-content/EN/TXT/PDF/?uri=CELEX:32006D0213

[4] https://huggingface.co/datasets/multi_eurlex/

we add the labels for each document by using the dataset by Chalkidis et al. (2021). In other words, the texts and bounding boxes come from Step 3 above and the images come from Step 2 above. We only reuse the CELEX ID and labels from the dataset by Chalkidis et al. (2021). Since we applied OCR and directly extracted texts from the original source PDF documents and converted PDF documents into images, any structured elements like tables and figures are retained in our curated dataset unlike the dataset created by Chalkidis et al. (2021) where the HTML mark-up texts are used instead of PDFs.[5]

**Rendered Wikipedia Articles (WIKI-DOC)** We additionally curate the WIKI-DOC dataset which complements MULTIEURLEX-DOC in the following ways: (1) It contains documents other than legal domain, and (2) It contains documents in non-European languages written in scripts other than Latin. Most Document AI models are OCR-dependent, and hence they suffer from error propagation due to incorrect OCR. This is especially an issue for languages *not* written in Latin or Cyrillic scripts since such languages are reported to have fewer OCR errors (Ignat et al., 2022). An example is Arabic, a language with right-to-left writing, which results in higher OCR error rate (Hegghammer, 2021). To complement the language and domain coverage in MULTIEURLEX-DOC, we use the documents of the dataset created by Sinha et al. (2018) by scraping and rendering Wikipedia articles. See Appendix B.1 for the full details on the creation steps.

## 4 Experiments

We now empirically explore both intra- and cross-lingual generalization ability of Document AI models for document classification for multiple languages. We ask the following research questions:

**RQ1:** Do Document AI models, specifically those that are multilingual in nature, perform equally accurately across languages?

**RQ2:** Can Document AI models classify documents in a cross-lingual transfer setting where we train on English and evaluate on other languages?

We conduct experiments under two different settings to answer these questions: 1) intralingual

[5]https://huggingface.co/datasets/multi_eurlex#source-data

| Model | T | L | I | Langs | Pretrain Doc. Images |
|---|---|---|---|---|---|
| InfoXLM | ✓ | | | 100 | - |
| LiLT | ✓ | ✓ | | 100 | IIT-CDIP |
| LayoutXLM | ✓ | ✓ | ✓ | 53 | IIT-CDIP + CC |
| Docformer | ✓ | ✓ | ✓ | 1 | IIT-CDIP |
| Donut | | | ✓ | 4 | IIT-CDIP + Syn. Wiki |

Table 2: Overview of the Document AI models used in this work with their corresponding input modalities (T: text, L: layout, I: image). Most models are pretrained on the English-only IIT-CDIP dataset (Lewis et al., 2006). LayoutXLM is also pretrained on documents from Common Crawl (CC) and Donut on synthetic Wikipedia documents (Syn. Wiki). InfoXLM is not pretrained on document images.

setup, where we train and evaluate on the same language, and 2) cross-lingual setup, where we train and evaluate on different languages.

In the intralingual setting, we are interested in the accuracy difference between multi-modal models and uni-modal (i.e., text- or image-only) models. Specifically, we are interested in knowing whether multi-modal models achieve higher accuracy in content classification datasets. Kim et al. (2022) report that the image-only model is more accurate than a multi-modal model on RVL-CDIP, but it is unclear if this will hold true for the datasets introduced in this work, as they require a deeper understanding of the text.

In the cross-lingual setting, we are interested in the cross-lingual generalization ability of Document AI models. We experiment on the *(zero-shot) cross-lingual transfer* setting, where we only have English training and validation documents. This is a practical setting since labeled document images are often more scarce than plain texts, especially in non-English languages. Both uni-modal and multi-modal have potential strengths in the cross-lingual transfer setting. On one hand, multi-modal models consume various input signals that are expected to transfer across certain languages (e.g., the layout information as reported by Wang et al. (2022)). On the other hand, image-only models can be applied to languages even unseen during pretraining as they are not dependent on the vocabulary of the model.

### 4.1 Experimental Setup

#### 4.1.1 Models

We now describe the Document AI models we experiment with in this paper. Our focus is on experimenting with diverse models that consume different input modalities (text, and/or document

layout, and/or images) and models that support multiple languages (Table 2). We choose representative model candidates for different settings. In all cases, we use the base model unless specified.

**InfoXLM (Chi et al., 2021) (Text-only)** InfoXLM is a text-only RoBERTa-like model which uses the same architecture as XLM-R (Conneau et al., 2020). We select InfoXLM instead of other text-only models following the prior work by Wang et al. (2022).

**LiLT + InfoXLM (Wang et al., 2022) (Text + Layout)** LiLT uses two different Transformers, one dedicated to text (initialized with a multilingual Transformer model checkpoint) and another for layout, allowing for plug-and-play of arbitrary text-only models with the same architecture while keeping the layout Transformers part of the model. As a result if initialized with a multi-lingual text model (such as XLM-R), LiLT can be pre-trained with only English documents but fine-tuned on any language. We follow Wang et al. (2022) and use InfoXLM as the underlying text model.

**DocFormer (Appalaraju et al., 2021) (Text + layout + image)** DocFormer is an encoder-only Transformer model with a CNN for visual feature extraction. It uses multi-modal self-attention to fuse visual and layout features at every layer to enforce an information residual connection to learn better cross-modal feature representations. We follow Appalaraju et al. (2021) and use the model with an attached linear classification layer.

**LayoutXLM (Xu et al., 2022) (Text + layout + image)** LayoutXLM uses the same architecture as LayoutLMv2 (Xu et al., 2021a), which is a multimodal Transformer model which extends LayoutLM (Xu et al., 2020) by adding the document image as an input to the model in addition to text and layout inputs. LayoutXLM is pretrained on PDFs from 53 languages extracted from Common Crawl[6], thus enabling it to process documents in multiple languages. Following multilingual pretraining convention, the data follows exponential sampling to handle the imbalance across languages.

**Donut (Kim et al., 2022) (Image-only)** Donut is an encoder-decoder model where the encoder is Swin Transformer (Liu et al., 2021) and the decoder is mBART (Liu et al., 2020). Donut only requires document images as inputs; this removes the dependency on OCR to extract text and layout information. Hence, Donut can be even applied to languages unseen during pretraining stage as it no longer requires a tokenizer. Except for Donut, all other models considered in the paper are encoder-only models. We experiment on an encoder-only version of Donut, where we remove the decoder layers from the original model and replace them with a single linear classification layer.[7] Donut is pretrained with an OCR-like task where the input is the document image and the previously decoded text and the output is the OCR output.

### 4.1.2 Hyperparameters

The hyperparameters of each model are tuned using English and one other non-English language, using the hyperparameters from the original papers as a recommendation. We use class-balanced training for all experiments in order to handle class imbalances. The chosen hyperparameters for each model are in Appendix C. All experiments are conducted on AWS p3.16xlarge instances with 8 V100 16GB memory GPUs.

### 4.1.3 Dataset Preprocessing

For both MULTIEURLEX-DOC and WIKI-DOC, we use pdf2image[8] to convert a PDF into its corresponding image using dpi of 300. We then use Tesseract 5.1 (Smith, 2007) to run OCR and extract words and bounding boxes. We use Transformers library (Wolf et al., 2020) for implementation. Finally, we only consider the first page of each PDF document, as similar truncation approaches have been shown to be strong baselines for text classification (Park et al., 2022). We leave long document classification for future work.

**MULTIEURLEX-DOC Preprocessing** Since MULTIEURLEX-DOC is a multi-label dataset, we convert the list of labels into multi-hot vectors. Refer to Appendix B (Table 11) for language specific data splits.

**WIKI-DOC Preprocessing** We make WIKI-DOC challenging for existing Document AI models by performing a series of preprocessing steps on

---

[6]https://commoncrawl.org/

[7]Our initial experiments on the encoder-decoder version on English WIKI-DOC in 10-shot setting scored macro F1 of $9.27_{2.92}$ which is significantly lower than the encoder-only version which scored $26.63_{3.06}$. Therefore, we focus on the encoder-only version.

[8]https://pypi.org/project/pdf2image/

| Models | Eurlex | Wiki |
|--------|--------|------|
| InfoXLM | $64.98_{1.7}$ | $94.04_{0.17}$ |
| LiLT | $61.56_{2.6}$ | $94.15_{0.11}$ |
| LayoutXLM | $65.67_{0.5}$ | $94.40_{0.13}$ |
| Donut | $45.27_{3.3}$ | $90.13_{0.58}$ |
| DocFormer | $63.46_{0.6}$ | $94.77_{0.10}$ |

Table 3: English results (en→ en) on MULTIEURLEX-DOC (Eurlex) and WIKI-DOC (Wiki).

the curated dataset: 1) **Few-shot Setting**: We subsample each class in the training split to be 10 training examples per class and created 5 different splits with different random seeds. This setting aims to test how much models learn from few examples in contrast to the RVL-CDIP dataset which includes 25K examples per class. 2) We further merge the subset of 219 classes curated by Sinha et al. (2018), which scored higher class-wise F1 scores than a threshold[9], into a single "Other" class. The Wikipedia language links are used to retrieve and curate the corresponding article in other languages. We further filter and keep Wikipedia articles which only use images with research-permissible licenses. The resulting dataset statistics after filtering and preprocessing are shown in Appendix B (Table 10).

**Evaluation Metrics**  For multi-label classification, we use mean R-Precision (mRP) as the evaluation metric following Chalkidis et al. (2021). To compute mRP, we rank the predicted labels in decreasing order of the confidence scores and compute Precision@$k$ where $k$ is the number of gold labels for the given document. For each model and language pair, we report the average mRP and standard deviation across 3 different runs. For multiclass classification, we report the average macro F1 scores and standard deviations.

## 4.2  English Results

We first experiment on the English portion of MULTIEURLEX-DOC and WIKI-DOC to confirm whether multilingual models are competitive with the English model scoring high accuracy on RVL-CDIP i.e., DocFormer (Appalaraju et al., 2021).

Results in Table 3 confirm that InfoXLM and LayoutXLM yield very similar results on English compared to DocFormer, and in fact are slightly more accurate on MULTIEURLEX-DOC. On the other hand, the accuracy of the OCR-free Donut model in English is relatively low. This is unlike

9set to 0.8 based on the development set

what is reported by Kim et al. (2022) where Donut is reported to be better than the multi-modal model (i.e., LayoutLM v2) on the RVL-CDIP dataset. To further analyze these results, we compare Donut and LayoutXLM under the few-shot setting described in Section 4.1.3. We observe that the macro F1 gap between the two models is significantly smaller in the full-shot setting (90.13 vs. 94.40) than the few-shot setting (26.63 vs. 82.18, full results at Appendix E.1). Thus, having a large number of finetuning examples is crucial for obtaining high accuracy using Donut.

Given the results on these two datasets, we focus on the four models pretrained on multiple languages in the remaining experiments to study the cross-lingual transferability.

## 4.3  Non-English Intralingual Results

### 4.3.1  MULTIEURLEX-DOC Results

Next, we evaluate the models on other languages, starting with MULTIEURLEX-DOC. The average mRP scores averaged across languages in Table 4 show that LayoutXLM is the most accurate. Both InfoXLM and LiLT perform poorly on certain languages either due to low score or high variance whereas we find LayoutXLM to be relatively consistent in its performance across languages. On the other hand, the encoder only Donut results are significantly lower (36.64), which can likely be attributed to the fact that the multi-label classification task requires identifying certain spans or distribution of words that indicate a specific attribute/label of the document. An image-only model like Donut is expected to struggle in capturing such nuances from the visual document structure and correlate them to a group of labels describing the document contents rather than document types, without the knowledge of the words comprising it.

We can also break down the results in Table 4 by language groups: Germanic (da, de, nl, sv), Romance (ro, es, fr, it, pt), Slavic (pl, bg, cs), and Uralic (hu, fi, et). InfoXLM and LayoutXLM have similar mRP on Germanic and Romance languages except Romanian (ro). However, LayoutXLM is scoring higher accuracy on average than InfoXLM on Slavic (hu: 63.87 vs. 58.84, fi: 63.52 vs. 63.46, et: 63.43 vs. 60.37) and Uralic (pl: 63.26 vs. 61.12, bg: 63.67 vs. 14.23, cs: 63.60 vs. 40.99) languages.

We do not see a strong correlation between the amount of training data in a given language and

| Models | da | de | nl | sv | ro | es | fr | it | pt |
|---|---|---|---|---|---|---|---|---|---|
| InfoXLM | $63.16_{1.2}$ | $63.89_{0.9}$ | $62.82_{3.6}$ | $64.08_{1.1}$ | $28.31_{24.7}$ | $63.2_{1.7}$ | $65.12_{0.5}$ | $64.74_{1.4}$ | $64.01_{1.5}$ |
| LiLT | $42.57_{28.9}$ | $61.48_{1.3}$ | $59.14_{2.9}$ | $63.78_{0.5}$ | $1.01_{0.3}$ | $63.1_{1.7}$ | $42.04_{30.6}$ | $62.84_{0.7}$ | $58.10_{2.4}$ |
| LayoutXLM | $65.17_{0.7}$ | $65.09_{0.5}$ | $65.07_{0.2}$ | $64.76_{1.0}$ | $64.15_{1.1}$ | $65.25_{0.3}$ | $65.36_{0.7}$ | $65.22_{0.3}$ | $64.26_{0.2}$ |
| Donut | $39.22_{6.9}$ | $40.37_{5.8}$ | $40.48_{1.6}$ | $35.53_{4.7}$ | $26.10_{0.6}$ | $34.99_{5.6}$ | $41.83_{3.4}$ | $41.32_{5.4}$ | $40.18_{7.7}$ |

| Models | pl | bg | cs | hu | fi | el | et | Avg |
|---|---|---|---|---|---|---|---|---|
| InfoXLM | $61.12_{0.8}$ | $14.23_{0.1}$ | $40.99_{34.9}$ | $58.84_{0.9}$ | $63.46_{1.0}$ | $63.95_{0.7}$ | $60.37_{0.8}$ | 56.89 |
| LiLT | $58.85_{0.5}$ | $1.55_{2.1}$ | $37.60_{31.8}$ | $39.27_{33.8}$ | $61.75_{0.6}$ | $60.45_{1.8}$ | $59.26_{0.9}$ | 49.08 |
| LayoutXLM | $63.26_{0.7}$ | $63.67_{0.6}$ | $63.6_{0.3}$ | $63.87_{0.8}$ | $63.52_{1.1}$ | $62.19_{0.3}$ | $63.43_{0.2}$ | 64.32 |
| Donut | $33.26_{2.7}$ | $27.85_{1.7}$ | $32.24_{0.1}$ | $34.03_{3.7}$ | $31.92_{1.2}$ | $43.56_{0.1}$ | $34.83_{0.5}$ | 36.64 |

Table 4: Intralingual results (X → X) on MULTIEURLEX-DOC at level 3. We report the average and standard deviation of mRP scores across 3 different random seeds. "Avg" indicates average across all languages.

| Models | es | fr | it | de | pt | zh | ja | ar |
|---|---|---|---|---|---|---|---|---|
| *Few-shot Setting* | | | | | | | | |
| InfoXLM | $77.97_{0.91}$ | $77.33_{0.52}$ | $78.28_{1.46}$ | $78.10_{0.89}$ | $76.84_{1.49}$ | $72.93_{0.83}$ | $74.97_{4.43}$ | $76.11_{1.94}$ |
| LiLT | $77.21_{1.12}$ | $76.24_{0.48}$ | $76.17_{1.57}$ | $76.84_{0.97}$ | $75.90_{0.45}$ | $70.69_{2.64}$ | $75.41_{0.73}$ | $75.05_{2.47}$ |
| LayoutXLM | $71.89_{7.29}$ | $77.82_{1.63}$ | $64.95_{6.70}$ | $75.85_{2.18}$ | $76.63_{0.96}$ | $65.31_{4.79}$ | $70.16_{3.06}$ | $60.49_{4.09}$ |
| Donut | $24.85_{1.88}$ | $32.77_{5.76}$ | $36.95_{2.49}$ | $21.81_{3.02}$ | $27.50_{3.42}$ | $24.07_{1.43}$ | $28.73_{2.25}$ | $38.71_{3.82}$ |
| *Full-shot Setting* | | | | | | | | |
| InfoXLM | △11.73 | △11.97 | △9.25 | △11.06 | △11.54 | △12.37 | △12.88 | △11.03 |
| LiLT | △12.28 | △11.89 | △11.97 | △11.85 | △10.53 | △15.89 | △11.98 | △11.47 |
| LayoutXLM | △16.52 | △10.52 | △23.52 | △13.16 | △8.98 | △19.94 | △15.98 | △25.05 |
| Donut | △42.52 | △31.45 | △28.24 | △34.32 | △26.98 | △38.95 | △30.97 | △21.94 |

Table 5: Few- and full-shot intralingual results (X → X) on WIKI-DOC. We report the average and standard deviation of macro F1 scores across 5 different random shots. For the full-shot setting, we report the score gap between the few-shot setting. The largest score gap between full- and few-shot setting is Arabic (ar) for LayoutXLM.

LayoutXLM's (model with best intra-lingual results) intra-lingual accuracy. For Greek (el), there are approximately 55K training examples (see Appendix B Table 11) but we see lowest mRP score of 62.19. On the contrary, there are languages such as Romanian and Bulgarian with approximately 16K training examples for which the mRP score is 64.15 and 63.67 respectively. For InfoXLM, we do see some correlation between the amount of training data and intra-lingual performance across languages. We conjecture that it is tied to the fact that LayoutXLM has been pre-trained with layout information so it is able to perform well even with less amount of data whereas InfoXLM being a text only model needs more data for the task.

### 4.3.2 WIKI-DOC Results

We now turn to experimenting on WIKI-DOC which includes non-European languages such as Chinese, Japanese, and Arabic (Table 5). Focusing on the few-shot LayoutXLM results, the largest accuracy gap with other models is on Arabic (< 15 F1 point between InfoXLM). We hypothesize that this is due to multiple factors. First, Arabic has a higher OCR error rate than languages that use Latin scripts (Hegghammer, 2021). Second, the Arabic pretraining data is relatively smaller than the other eight languages when pretraining LayoutXLM (Xu et al., 2022). Lastly, the layout position of Arabic texts are reversed unlike the other eight languages, making it harder for LayoutXLM to learn 2D position embeddings during pretraining. Figure 2 further shows that semantically close classes are often misclassified by LayoutXLM such as "Fish" vs. "Amphibian" and "Mollusca" vs. "Crustacean".

In contrast, Donut scores are significantly lower than other models (Table 5). This is likely due to Donut being pretrained only on English with real PDF documents (i.e., IIT-CDIP) and on synthetic Chinese, Japanese, and Korean documents (Kim et al., 2022). Also, models consuming images (i.e., Donut and LayoutXLM) are not the most accurate, especially when the input text contains strong signals for classifying documents.[10]

---

[10]We didn't include other image-only models like Document Image Transformer (Li et al., 2022b, DiT) since Donut was seen to show much stronger performance than DiT on RVL-CDIP dataset (95.30 for Donut (Kim et al., 2022) vs

| Models | da | de | nl | sv | ro | es | fr | it | pt |
|---|---|---|---|---|---|---|---|---|---|
| InfoXLM | $51.28_{1.3}$ | $53.27_{1.3}$ | $46.47_{1.4}$ | $47.91_{2.5}$ | $48.73_{2.8}$ | $52.63_{2.4}$ | $52.25_{2.6}$ | $47.75_{2.3}$ | $47.98_{1.0}$ |
| LiLT | $43.94_{6.1}$ | $44.30_{5.8}$ | $38.75_{5.4}$ | $42.11_{7.3}$ | $43.32_{5.7}$ | $47.41_{4.6}$ | $43.96_{6.3}$ | $45.43_{3.4}$ | $42.99_{5.5}$ |
| LayoutXLM | $51.29_{1.7}$ | $46.26_{1.5}$ | $46.49_{2.9}$ | $47.75_{1.5}$ | $50.15_{2.1}$ | $52.35_{1.1}$ | $52.50_{0.7}$ | $49.33_{1.6}$ | $48.46_{1.7}$ |
| Donut | $16.97_{2.4}$ | $14.08_{0.7}$ | $16.68_{0.9}$ | $18.13_{2.8}$ | $21.41_{2.7}$ | $18.55_{1.5}$ | $19.02_{2.4}$ | $18.36_{1.1}$ | $20.00_{2.1}$ |

| Models | pl | bg | cs | hu | fi | el | et | Avg |
|---|---|---|---|---|---|---|---|---|
| InfoXLM | $41.62_{0.6}$ | $45.78_{2.3}$ | $46.35_{2.2}$ | $45.74_{3.4}$ | $42.86_{3.4}$ | $34.87_{2.4}$ | $41.78_{3.7}$ | 46.70 |
| LiLT | $35.49_{5.3}$ | $40.77_{6.9}$ | $37.28_{8.0}$ | $39.03_{6.8}$ | $34.02_{7.0}$ | $27.17_{4.1}$ | $34.41_{7.1}$ | 40.02 |
| LayoutXLM | $41.28_{2.7}$ | $47.31_{1.3}$ | $42.32_{2.2}$ | $39.36_{0.9}$ | $31.85_{1.5}$ | $27.15_{1.4}$ | $38.37_{1.8}$ | 44.51 |
| Donut | $11.45_{2.6}$ | $6.58_{0.6}$ | $12.94_{2.8}$ | $7.53_{0.5}$ | $9.39_{2.3}$ | $5.56_{0.9}$ | $15.16_{2.6}$ | 14.49 |

Table 6: Cross-lingual results (en → X) on MULTIEURLEX-DOC at level 3. We report the average and standard deviation of mRP scores across 3 different random seeds. "Avg" indicates average across all languages.

| Models | es | fr | it | de | pt | zh | ja | ar |
|---|---|---|---|---|---|---|---|---|
| *Few-shot Setting* | | | | | | | | |
| InfoXLM | $59.36_{1.07}$ | $60.37_{0.94}$ | $50.17_{1.75}$ | $59.17_{1.11}$ | $58.96_{1.01}$ | $44.26_{0.86}$ | $39.05_{1.09}$ | $39.30_{1.80}$ |
| LiLT | $60.16_{1.03}$ | $59.19_{1.39}$ | $49.73_{1.72}$ | $59.14_{1.10}$ | $57.82_{0.89}$ | $44.59_{0.85}$ | $39.57_{1.61}$ | $38.23_{1.96}$ |
| LayoutXLM | $49.73_{4.38}$ | $48.31_{5.36}$ | $42.07_{5.83}$ | $47.34_{5.48}$ | $46.89_{5.14}$ | $29.21_{4.29}$ | $27.65_{5.82}$ | $24.04_{5.98}$ |
| Donut | $4.70_{1.09}$ | $3.45_{1.02}$ | $4.65_{0.64}$ | $5.91_{1.63}$ | $3.75_{1.18}$ | $2.26_{0.26}$ | $2.50_{0.75}$ | $2.37_{0.83}$ |
| *Full-shot Setting* | | | | | | | | |
| InfoXLM | △13.60 | △10.77 | △18.44 | △11.87 | △11.30 | △8.12 | △6.28 | △7.79 |
| LiLT | △13.90 | △11.98 | △16.82 | △10.63 | △13.04 | △9.86 | △7.14 | △5.95 |
| LayoutXLM | △9.26 | △10.23 | △11.75 | △7.26 | △9.17 | △0.03 | ▼5.39 | ▼3.04 |
| Donut | △3.20 | △3.40 | △3.94 | △3.01 | △7.24 | △5.18 | △0.52 | ▼0.05 |

Table 7: Cross-lingual transfer Macro F1 results (en → X) on WIKI-DOC with full- and few-shot (10-shot) setup. We report the accuracy difference between few-shot and full-shot setting. The scores either stays on par or decrease in Chinese, Japanese, and Arabic for LayoutXLM and Donut when comparing the full and few-shot (10-shot) setup.

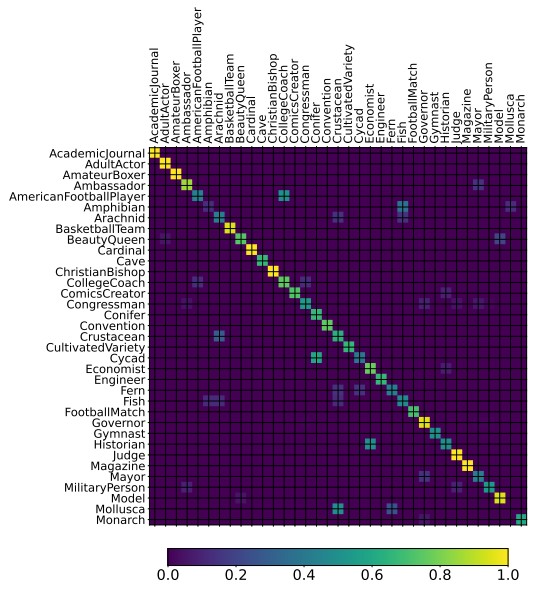

Figure 2: The confusion matrix of the LayoutXLM prediction results on the first 35 classes in Arabic WIKI-DOC under intralingual setting. Semantically close classes (e.g., "Mollusca" vs. "Crustacean") are often misclassified, challenging LayoutXLM on this dataset.

### 4.4 Cross-Lingual Experiments

We now explore the cross-lingual generalization ability of Document AI models by evaluating the models in cross-lingual transfer setting, i.e., fine-tuning only on English and directly evaluating on non-English target languages (en → X).

#### 4.4.1 MULTIEURLEX-DOC Results

In Table 6, we observe that even though LayoutXLM performs the best on individual languages in intralingual setting, it does not generalize as well as InfoXLM in cross-lingual setting. This result is a bit surprising because the parallel documents across languages in the dataset have the same layout information thus the layout and image features should be easily transferable across languages.

Across language groups, we can see that both InfoXLM and LayoutXLM perform similarly on Germanic languages (da, nl, sv) except German (de) where InfoXLM does much better. On the other hand, LayoutXLM performs slightly better (than InfoXLM) on Romance languages (ro, es, fr, it, pt). However most of the accuracy gap between the two models is introduced due to Uralic lan-

---

92.69 for DiT-Large (Li et al., 2022b)) and publicly available DiT checkpoints are not pretrained in multiple languages.

guages (hu, fi, et) where InfoXLM yields far better results.

### 4.4.2 WIKI-DOC Results

From Table 7, we also see that Donut cannot generalize across languages. These and other results from this section indicate that there are large areas of improvement on either the model or the pretraining strategy for Donut to generalize across langauges. Finally, Table 7 also shows the limited cross-lingual transferability due to the accuracy drop for LayoutXLM in Japanese (▼5.39) and Arabic (▼3.04) when increasing the number of English training examples from few-shot (10) to full-shot setup. We further analyze the correlation to typological features in the next section.

### 4.5 Correlation with Typological Features

To further understand the cross-lingual transfer results in Tables 6 and 7, we look at the correlation between these results and the typological features (syntactic, phonological, and phonetic inventory features) of the languages involved. A higher correlation implies that the cross-lingual gap is harder to bridge via those features. Inspired from Lauscher et al. (2020), we analyze the correlation of cross-lingual transfer gap with the typological distance between the source and target languages. Because the test sets of the newly curated datasets are not completely parallel across all languages, we measure the accuracy gap between models trained on source and target languages, and evaluate those on the same target language test set. Specifically, the correlation is calculated among the two set of numbers. The first set is the accuracy gap between a model trained on English and evaluated on language X (en → X) and a model trained on language X and evaluated on language X (en → X). The second set is the typological distance between English and the target languages where we use the precomputed typological distance between languages from LANG2VEC (Littell et al., 2017).

The correlation analysis results (Table 8) show that the transfer gap is highly correlated with the syntactic cosine distance between the source and the target language. This further explains the gap between few-shot and full-shot cross-lingual transfer results in Table 7 where increasing the training examples in the source language (i.e., English) hurts the accuracy of LayoutXLM in Japanese and Arabic, which have the highest syntactic distance from English (ja: .66, ar: .57) among the 8 lan-

| Data | Model | SYN | | PHON | | INV | |
|------|-------|-----|-----|------|-----|-----|-----|
| | | P | S | P | S | P | S |
| Eurlex | InfoXLM | .44 | **.65** | -.36 | -.27 | .23 | .21 |
| | LayoutXLM | .56 | **.68** | -.60 | -.52 | .07 | .06 |
| Wiki | InfoXLM | .91 | **.96** | .32 | .44 | .71 | .52 |
| | LayoutXLM | **.88** | **.88** | .27 | .34 | .75 | .57 |

Table 8: Correlation analysis on the cross-lingual transfer gap and typological distances using syntactic (SYN), phonological (PHON), and phonetic inventory (INV) features. Spearman (S) and Pearson (P) correlations are used. The highest correlations are in bold.

guages in WIKI-DOC. Similar trends are observed on MULTIEURLEX-DOC in Table 6 when comparing LayoutXLM and InfoXLM for the most syntactically-distant languages (cs: .66, hu: .60) among the 16 languages.

## 5 Conclusion

In this paper, we curated two new multilingual document image classification datasets, MULTIEURLEX-DOC and WIKI-DOC, to evaluate both the multilingual and cross-lingual generalization ability of Document AI models. Through benchmarking on the two newly curated datasets, we show strong intralingual results across languages of the multimodel model but also show the limited cross-lingual generalization ability of the multimodal model. Furthermore, the OCR-free or image-only model showed the largest gap between the best performing models, showing large areas of improvement in datasets which require deeper content understanding from texts.

Future work in this space should investigate improvement strategies of multi-modal and OCR-free Document AI models to enable them to achieve a deeper understanding of the text from document images. Finally, the curated datasets still cover only a small subset of languages spoken in the world. Future work should expand the datasets and experiments to more diverse set of language families.

## Acknowledgements

We sincerely thank the anonymous reviewers and the colleagues at AWS AI Labs for giving constructive feedback on the earlier versions of this paper. We also thank Philipp Schmid for the detailed and educative tutorial and the associated codes to experiment with Document AI models.[11]

---

[11] https://www.philschmid.de/fine-tuning-donut and https://www.philschmid.de/fine-tuning-lilt

## Limitations

**Low-Resource Language Coverage** The language covered by the two newly curated datasets is limited in terms of the coverage of language families (e.g., Afro-Asiatic family is not covered) especially on low-resource languages. We introduce WIKI-DOC to extend the language coverage beyond European languages which are covered by MULTIEURLEX-DOC but the dataset size becomes too small due to frequent missing inter-language links in Wikipedia and we have not covered low-resource languages in the newly curated dataset.

**English as the Source Language** The cross-lingual experiments conducted in the paper uses English as the source language to train the model. While it's true that choice of source language changes the downstream task results significantly (e.g., Lin et al. (2019)), we choose English as the source language from practical perspective and leave exploration of the choice of source languages in Document AI models as future work.

**Discrepancy in Pretraining Data among Models** We use the pretrained Document AI models out-of-the-box without any additional pretraining. As a result, there are slight discrepancies in the corpus used for pre-training each model (Table 2).

**Task Coverage** Our curated dataset is specifically designed for multi-class and multi-label document classification in multiple languages. To the best of our knowledge, XFUND (Xu et al., 2022) and EPHOIE (Wang et al., 2021) are the only publicly available non-English datasets to evaluate Document AI models. We therefore encourage the research community to build diverse set of document image datasets to cover various tasks in multiple languages.

**Diverse Document Layouts** The document layouts in our newly curated dataset are mostly static except for MULTIEURLEX-DOC being multi-column and some layout variation based on Wikipedia templates in WIKI-DOC. Layout-aware models tend to have the issue of layout distribution shifts (Chen et al., 2023a) and such issues may not be captured in the newly curated datasets.

**Larger Models** The size of all models benchmarked in this paper are $< 400M$ (Appendix A) and relatively smaller compared to models explored in the recent literature. Though not focused on document image classification, we leave it to other work (e.g., (Chen et al., 2023b)) and encourage further research on this topic by the community.

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

## A  Model Details

Table 9 shows the licenses of the publicly available checkpoints of the models used in this paper. Since LayoutXLM follows Attribution-NonCommercial-ShareAlike 4.0 International (CC BY-NC-SA 4.0) license, we solely used it for conducting experiments in this paper.

| Model | #Params | License |
|---|---|---|
| InfoXLM | 278M | MIT |
| LiLT | 284M | MIT |
| LayoutXLM | 369M | CC-BY-NC-SA 4.0 |
| Donut | 74M | MIT |

Table 9: The number of parameters and the license of the model explored in this paper.

## B  Dataset Details

Tables 10 and 11 show the dataset statistics after all the preprocessing steps for WIKI-DOC and MULTIEURLEX-DOC respectively. For MULTIEURLEX-DOC, we found that there are documents in the train/dev/test sets where we ran into errors when converting the pdf page into an image for OCR. Thus the number of documents in dev/test sets is slightly lower than 5k.

### B.1  WIKI-DOC Creation Steps

For curating documents for WIKI-DOC we conducted following steps:

1. Extract the document titles document labels used in Sinha et al. (2018).

2. Use the titles of the documents to retrieve English Wikipedia article page (if it gives redirection, discard that example).

| Lang. | #Cls | Train Full | Few | Val | Test |
|---|---|---|---|---|---|
| English (en) | 110 | 152K | 2K | 32K | 32K |
| German (de) | 103 | 41K | 1K | 8K | 8K |
| French (fr) | 101 | 33K | 1K | 7K | 7K |
| Spanish (es) | 106 | 42K | 1K | 9K | 9K |
| Portuguese (pt) | 86 | 33K | 1K | 4K | 4K |
| Italian (it) | 62 | 20K | 1K | 4K | 4K |
| Chinese (zh) | 90 | 23K | 1K | 4K | 4K |
| Japanese (ja) | 94 | 23K | 1K | 4K | 4K |
| Arabic (ar) | 60 | 8K | 1K | 1K | 1K |

Table 10: Statistics of WIKI-DOC on the number of classes (#Cls) examples before (Full) and after (Few) subsampling the training split to 10-shots.

| Lang. | Number of Documents (train/dev/test) |
|---|---|
| English (en) | 54808 / 4997 / 4988 |
| German (de) | 54804 / 4997 / 4988 |
| French (fr) | 54804 / 4997 / 4988 |
| Italian (it) | 54805 / 4997 / 4987 |
| Spanish (es) | 52621 / 4997 / 4988 |
| Polish (pl) | 23063 / 4997 / 4988 |
| Romanian (ro) | 15914 / 4997 / 4988 |
| Dutch (nl) | 54803 / 4997 / 4988 |
| Greek (el) | 54828 / 4997 / 4988 |
| Hungarian (hu) | 22542 / 4997 / 4988 |
| Portuguese (pt) | 52205 / 4997 / 4988 |
| Czech (cs) | 23056 / 4997 / 4988 |
| Swedish (sv) | 42356 / 4997 / 4988 |
| Bulgarian (bg) | 15979 / 4997 / 4988 |
| Danish (da) | 54806 / 4997 / 4988 |
| Finnish (fi) | 42362 / 4997 / 4988 |
| Slovak (sk) | 22858 / 4997 / 4988 |
| Lithuanian (lt) | 23075 / 4997 / 4987 |
| Croatian (hr) | 7944 / 2499 / 4988 |
| Slovene (sl) | 23061 / 4997 / 4988 |
| Estonian (et) | 22986 / 4997 / 4988 |
| Latvian (lv) | 23045 / 4997 / 4988 |
| Maltese (mt) | 17390 / 4996 / 4988 |

Table 11: Statistics of MULTIEURLEX-DOC dataset for different languages

3. Use Wikipedia inter-language links to retrieve non-English articles.

4. Since the licenses of each image used in Wikipedia articles differ, we use the Wikimedia API[12] to obtain the license information of each image and filter out the article if either the license is not available or it is not permissible for research purposes.

5. The filtered Wikipedia articles are converted from HTML to PDFs using pdfkit.[13]

Table 12 shows the licenses of the datasets we have built upon.

### B.2  Preprocessing Details

Languages with Latin script are processed with the language option of Tesseract set to the language of interest. For languages that use non-Latin scripts

---

[12]https://www.mediawiki.org/wiki/API:Imageinfo
[13]https://pypi.org/project/pdfkit/

| Dataset | License |
|---|---|
| Multi-Eurlex | CC BY-SA 4.0 |
| DBpedia | CC-BY-SA 3.0 |
| Wikipedia texts | CC-BY-SA 3.0 |
| Wikipedia images | Varies |

Table 12: The licenses of the datasets we have curated from and built upon. Most datasets follow Creative Commons Attribution-ShareAlike (CC-BY-SA) License.

such as Japanese, Chinese, and Arabic, we further include English as the sub-language model to conduct OCR on Latin texts appearing in non-Latin languages. For WIKI-DOC, it is split into train:validation:test to 7:1.5:1.5 following stratified sampling.

## C  Hyperparameters

Table 13 shows the hyperparameter used for evaluating on WIKI-DOC. We conduct manual search starting from the original hyperparameters reported in the original paper with the range of $[5e-6, 1e-4]$ for the learning rate and $[0, 100, 200]$ for the warm-up steps.

Similarly, table 14 covers the batch size, learning rate and warmup ratio used to train different models for MULTIEURLEX-DOC dataset.

| Model | Batch | LR | Warm-up |
|---|---|---|---|
| InfoXLM | 64 | 1e-5 | 200 steps |
| LiLT | 64 | 1e-5 | 200 steps |
| LayoutXLM | 32 | 2e-5 | 200 steps |
| Donut | 32 | 1e-4 | 200 steps |

Table 13: Selected hyperparameters for WIKI-DOC dataset. Learning rate (LR). Batch size is `per_device_train_batch_size` × the number of GPUs used during training.

## D  List of WIKI-DOC Classes

Table 16 shows the list of classes in the WIKI-DOC dataset after conducting preprocessing steps explained in Section 3.1. Some classes are specific to some countries (CanadianFootballTeam, EurovisionSongContestEntry) potentially encouraging the models to bias on such countries.

| Model | Batch | LR | Warm-up |
|---|---|---|---|
| InfoXLM | 64 | 3e-5 | 10% |
| LiLT | 64 | 3e-5 | 10% |
| LayoutXLM | 32 | 2e-5 | 10% |
| Donut | 32 | 1e-4 | 1000 steps |

Table 14: Selected hyperparameters for MULTIEURLEX-DOC dataset. Learning rate (LR). Batch size is `per_device_train_batch_size` × the number of GPUs used during training.

## E  Full Results

### E.1  Few- and Full-Shot Results on English WIKI-DOC

Table 15 shows the results on comparing few- and full-shot results on English WIKI-DOC.

| Models | Few | Full |
|---|---|---|
| InfoXLM | $81.30_{0.85}$ | $94.04_{0.17}$ |
| LiLT | $81.44_{0.90}$ | $94.15_{0.11}$ |
| LayoutXLM | $82.18_{1.62}$ | $94.40_{0.13}$ |
| Donut | $26.63_{3.06}$ | $90.13_{0.58}$ |

Table 15: English results (en→ en) on few- and full-shot setup on WIKI-DOC.

### E.2  Initial All Page Results on MULTIEURLEX-DOC

In Table 17, we include BERT and XLM-R results on all pages. Additionally we also include XLM-R results on first page of the PDF document. All Page results are borrowed from (Chalkidis et al., 2021). We can see that as we go from all pages to just first page, there is a drop in the score across all languages indicating that text from other pages is important because the classes at level 3 are very fine-grained.

### E.3  Correlation Analysis on Other Typological Features

Table 18 shows the correlation analysis results on all typological features using the pre-computed typological distances from Littell et al. (2017).

| | | |
|---|---|---|
| AcademicJournal | EurovisionSongContestEntry | Poem |
| AdultActor | Fern | Poet |
| Album | FilmFestival | Pope |
| AmateurBoxer | Fish | President |
| Ambassador | FootballMatch | PrimeMinister |
| AmericanFootballPlayer | Glacier | PublicTransitSystem |
| Amphibian | GolfTournament | Racecourse |
| AnimangaCharacter | Governor | RadioHost |
| Anime | Gymnast | RadioStation |
| Arachnid | Historian | Religious |
| Baronet | IceHockeyLeague | Reptile |
| BasketballTeam | Insect | Restaurant |
| BeautyQueen | Journalist | Road |
| BroadcastNetwork | Judge | RoadTunnel |
| BusCompany | Lighthouse | RollerCoaster |
| BusinessPerson | Magazine | RugbyClub |
| CanadianFootballTeam | Mayor | RugbyLeague |
| Canal | Medician | Saint |
| Cardinal | MemberOfParliament | School |
| Cave | MilitaryPerson | ScreenWriter |
| ChristianBishop | Model | Senator |
| ClassicalMusicArtist | Mollusca | ShoppingMall |
| ClassicalMusicComposition | Monarch | Skater |
| CollegeCoach | Moss | SoccerLeague |
| Comedian | Mountain | SoccerManager |
| ComicsCreator | MountainPass | SoccerPlayer |
| Congressman | MountainRange | SoccerTournament |
| Conifer | MusicFestival | SportsTeamMember |
| Convention | Musical | SumoWrestler |
| Cricketer | MythologicalFigure | TelevisionStation |
| Crustacean | Newspaper | TennisTournament |
| CultivatedVariety | Noble | TradeUnion |
| Cycad | OfficeHolder | University |
| Dam | Other | Village |
| Economist | Philosopher | VoiceActor |
| Engineer | Photographer | Volcano |
| Entomologist | PlayboyPlaymate | WrestlingEvent |

Table 16: Classes in the WIKI-DOC dataset.

| Models | en | da | de | nl | sv | ro | es | fr | it |
|---|---|---|---|---|---|---|---|---|---|
| | | | | All Pages | | | | | |
| NATIVE-BERT | 67.7 | 65.5 | 68.4 | 66.7 | 68.5 | 68.5 | 67.6 | 67.4 | 67.9 |
| XLM-R | 67.4 | 66.7 | 67.5 | 67.3 | 66.5 | 66.4 | 67.8 | 67.2 | 67.4 |
| | | | | First Page only | | | | | |
| XLM-R | $64.63_{1.9}$ | $64.65_{0.9}$ | $65.71_{0.2}$ | $64.7_{1.1}$ | $64.53_{1.36}$ | $56.51_{2.3}$ | $65.02_{0.7}$ | $65.2_{0.7}$ | $64.95_{0.5}$ |
| Models | pt | pl | bg | cs | hu | fi | el | et | Avg |
| | | | | All Pages | | | | | |
| NATIVE-BERT | 67.4 | 67.2 | - | 66.7 | 67.7 | 67.8 | 67.8 | 66 | 67.2 |
| XLM-R | 67 | 65 | 66.1 | 66.7 | 65.5 | 66.5 | 65.8 | 65.7 | 66.61 |
| | | | | First Page only | | | | | |
| XLM-R | $65.9_{0.4}$ | $61.56_{1.0}$ | $59.15_{0.2}$ | $61.54_{0.2}$ | $61.51_{0.7}$ | $64.15_{1.7}$ | $63.94_{1.2}$ | $61.28_{1.1}$ | 63.23 |

Table 17: Intralingual results (X → X) of BERT and XLM-R on the MULTIEURLEX-DOC dataset at level 3. "Avg" in the table indicates average across all languages.

| Data | Model | SYN | | PHON | | INV | | GEO | | GEN | | FEA | |
|---|---|---|---|---|---|---|---|---|---|---|---|---|---|
| | | P | S | P | S | P | S | P | S | P | S | P | S |
| Eurlex | InfoXLM | .44 | .65 | -.36 | -.27 | .23 | .21 | .30 | .29 | .30 | .30 | .02 | .16 |
| | LayoutXLM | .56 | .68 | -.60 | -.52 | .07 | .06 | .42 | .49 | .29 | .26 | -.52 | -.04 |
| Wiki | InfoXLM | .91 | .96 | .32 | .44 | .71 | .52 | .88 | .73 | .51 | .82 | .85 | .82 |
| | LayoutXLM | .88 | .88 | .27 | .34 | .75 | .57 | .90 | .81 | .46 | .72 | .82 | .72 |

Table 18: Full correlation analysis results on the cross-lingual transfer gap and typological distances using syntactic (SYN), phonological (PHON), phonetic inventory (INV), genetic (GEN), and featural (FEA) features. Spearman (S) and Pearson (P) correlations are used.