# OpenReview forum: "A Multi-Modal Multilingual Benchmark for Document Image Classification"
_EMNLP/2023/Conference — EMNLP 2023 Findings_

### Official Review · Reviewer_VmrD · 2023-08-05

**Soundness:** 3

**Excitement:**

3: Ambivalent: It has merits (e.g., it reports state-of-the-art results, the idea is nice), but there are key weaknesses (e.g., it describes incremental work), and it can significantly benefit from another round of revision. However, I won't object to accepting it if my co-reviewers champion it.

**Paper Topic And Main Contributions:**

This paper introduces two new datasets, MULTIEURLEX-DOC and WIKI-DOC, to advance multilingual document image classification. The key contributions are:

1. MULTIEURLEX-DOC curates a multi-label legal document dataset in 23 European languages derived from MULTIEURLEX.

2. WIKI-DOC curates a multi-class Wikipedia document dataset in 9 languages, including non-European languages like Arabic, Chinese, and Japanese.

3. Several multilingual Document AI models are evaluated on the new datasets to analyze their multilingual and cross-lingual abilities. Limitations in cross-lingual transfer are revealed, especially between distant languages.

4. Significant performance gaps exist between image-only models like Donut and the best models, indicating opportunities for improvement in image-only approaches.

5. The new datasets enable more comprehensive analysis of multilingual Document AI models, complementing existing datasets.

6. Multilingual model limitations are revealed, particularly in cross-lingual transfer and for image-only models.

In summary, the key contributions are curating two new challenging multilingual document datasets and leveraging them to analyze and reveal limitations of current multilingual Document AI models. The datasets advance the field's ability to evaluate and improve multilingual document classification.

**Reasons To Accept:**

1. Introduces two valuable new multilingual datasets for document image classification, filling an important gap and enabling more comprehensive evaluation of models.

2. Provides extensive empirical analysis and benchmarking of state-of-the-art document AI models, shedding light on their capabilities and limitations especially for multilingual scenarios.

3. Compares uni-modal and multi-modal models in multilingual settings, finding image-only models struggle in certain scenarios unlike prior English-only results.

4. Open sources code and datasets to facilitate reproducible research.

**Reasons To Reject:**

1. Insufficient details and analysis of proposed datasets: More in-depth information about the proposed datasets, including label semantics, distribution, and discussion of the influence of domain for MultiEurLex-Doc and Wiki-Doc, is necessary. This additional context would clarify the dataset's strengths and limitations. For example, InfoXLM performs well on Wiki-Doc possibly due to its pre-training on Wikipedia.

2. The paper could benefit from including discussions and insights about vision transformer-based models, such as DiT, and comparing them with image-only vision encoders and end-to-end encoder-decoder models like Donut.

**Reproducibility:**

4: Could mostly reproduce the results, but there may be some variation because of sample variance or minor variations in their interpretation of the protocol or method.

**Reviewer Confidence:**

4: Quite sure. I tried to check the important points carefully. It's unlikely, though conceivable, that I missed something that should affect my ratings.

---

> ### Author Rebuttal · Authors · 2023-08-28
>
> We thank the reviewer for providing detailed feedback on our work.
>
> ### Rebuttal to Reasons To Reject
>
> > “Insufficient details and analysis of proposed datasets, including label semantics, distribution, and discussion of the influence of domain”
>
>
> We would like to remind that the dataset distribution details provided in Appendix B, Tables 10 and 11 of the paper.  Regarding the influence of domain, we agree that different models use different pre-training corpus (Table 2) and therefore would exhibit slight biases towards different data domains (e.g., less domain gap between InfoXLM on the WIKI-DOC dataset) and thus we have stated it as one of the limitations of the work. We will add additional insights while comparing the models performances based on these pre-training variations.
>
>
> > “could benefit from including discussions and insights about vision transformer-based models, such as DiT,  and comparing them with image-only vision encoders”
>
>
> We have included Donut as part of the image-based models for our benchmarking, however, we didn’t include other image-only models like Document Image Transformer (DiT) since Donut was seen to show much stronger performance than DiT on RVL-CDIP dataset (95.30 for Donut [1] vs 92.69 of the DiT-Large model [2]) and hence we considered Donut as a suitable representation for image-based models. Moreover, image-only models have typically under-performed compared to image+text models for document understanding tasks (Table 4 in [3]) and therefore we limited our analysis to one model i.e., Donut. Finally, the DiT models from [2] are pretrained only on English document images (i.e., IIT-CDIP dataset [3]), and therefore, we excluded from experimenting in our submission due to focus on experimenting in non-English languages.
>
> [1] OCR-free Document Understanding Transformer https://arxiv.org/abs/2111.15664
>
> [2] DiT: Self-supervised Pre-training for Document Image Transformer https://arxiv.org/abs/2203.02378
>
> [3] DocFormerv2: Local Features for Document Understanding https://arxiv.org/pdf/2306.01733.pdf
>
> [4] Building a test collection for complex document information processing  https://dl.acm.org/doi/10.1145/1148170.1148307

---

### Official Review · Reviewer_BLSd · 2023-08-05

**Soundness:** 3

**Excitement:**

3: Ambivalent: It has merits (e.g., it reports state-of-the-art results, the idea is nice), but there are key weaknesses (e.g., it describes incremental work), and it can significantly benefit from another round of revision. However, I won't object to accepting it if my co-reviewers champion it.

**Missing References:**

XFUND is mentioned in "Task coverage" section.  However, it is not clear why there are no comparisons or any discussion in the text about this multilingual dataset. It would be nice to see a comparison of especially the zero/few shot experiments with the ones in the LayoutXLM paper. Since the reference is already included because of the "Task coverage" section I am not including it again.

**Paper Topic And Main Contributions:**

The paper describes a document image classification benchmark. The authors introduce two multilingual datasets and perform experiments on these with out-of-the-box Document AI models, namely LayoutXLM, InfoLXM, Donut and LiLT. They also use DocFormer, which does not feature in all experiments.

The two multilingual datasets that are 'curated' are 1.MultiEurLex-Doc and 2.Wiki-Doc. These are both derived from previous work. The former has documents about EU laws in 23 different languages and the latter is created from Wikipedia articles by a previous study, and has non-latin script languages such as Arabic to complement MultiEurLex-Doc.

The paper aims to find answers to 3 research questions: 1.Whether pre-trained models for document classification work on their datasets. 2.To test how the multilingual aspect of the pre-trained models perform on different languages. 3.Cross-lingual aspect of these models, meaning how well they perform in zero-shot transfer environments where the training language is English and the model is tested on other languages.

The authors perform experiments where they test these pre-trained models that use different sets of modalities of 1.Document Layout, 2.Images, 3.Text. They report results for each language in the two datasets in the first round of results. Then, they report other results for cross-lingual transfer where the source language is English in all cases. They also report "full-shot" and "few-shot" results for all languages. Furthermore, they report challenges in cross-lingual transfer in syntactically distant languages with English, which seems logical. They also report that image-only models are not as successful as the multi-modal or text-based ones in these settings.

**Questions For The Authors:**

In addition to the issues mentioned in the reasons to reject section, here are a few more questions:

Ln. 270: What do you mean here? It is used because it was previously used? I was confused.  This is not a self-reference is it?

Ln 337.  There are doubts about the methodology here. Creating pdf’s from text (in MultiEURLEX) and then converting them to images with pdf2image is going to affect the real life performance, since the OCR output you get after putting these images through this pipeline will probably create much less noise. Have you considered this?

Ln 400: Where do we observe the result mentioned here? Is it referred to in the tables for few shot learning?

Ln 429: I did not find a table I can refer to for this information.
Table 8. Can you explain how the correlation is calculated more clearly please?

**Reasons To Accept:**

The authors present two multilingual datasets for the task in hand. They are sizeable datasets and, since they were created from existing datasets, they are likely to be cleaner than a brand new one.

There are a substantial number of multilingual experiments on document image classification.  They do intra-lingual and cross-lingual experiments on the datasets. The intra-lingual ones aim to test multi-modal vs image-only model performance comparison in any given languages by testing the pretrained out of the box models. The cross-lingual experiments aim to test few-shot learning on the cross-lingual transfer setting for a pair of languages where the model is trained on English and tested on a different language. They test uni-model and multi-modal models in this setting.

**Reasons To Reject:**

There are several issues about methodology and the data used in the paper:

A) It is not entirely clear how the two datasets are "curated" from existing datasets. The only mention of the modification I could find was that some files were removed, and text files were converted to PDF files in the MULTIEURLEX case. The original label sets were kept. Is there any other major alteration that I might have missed? If not, I would suggest not to call them “novel datasets”.

B) Regarding the methodology used, the authors first convert texts into pdf format, which was later converted to images for some models. Doesn’t this automatically cause some disadvantage for layout- and image-based models and an advantage for text-only models since that is the source of primary information for those models? I think the way the datasets were created and were tested on models that were  trained on radically different kinds of examples is the cause of many unexplained results the authors observed in the experiments such as InfoXML beating other models in some settings and Donut being very considerably worse than all other models in almost all settings.

C) Regarding the motivation for creating/curating these particular datasets, the authors claim the datasets complement the existing ones in some respects. In addition, these two datasets seem to be completely different in content/characteristics from the one the authors are comparing their work to for the task of Document AI. For instance, the category set in RVL-CDIP has only 16 categories whereas the law document dataset has hundreds, MultiEURLEX reportedly only has text documents.  I am wondering whether these datasets were chosen only because they were multilingual rather than whether they were appropriate for document image classification tasks or not.

**Reproducibility:**

4: Could mostly reproduce the results, but there may be some variation because of sample variance or minor variations in their interpretation of the protocol or method.

**Reviewer Confidence:**

5: Positive that my evaluation is correct. I read the paper very carefully and I am very familiar with related work.

---

> ### Author Rebuttal · Authors · 2023-08-28
>
> We thank the reviewer for providing detailed feedback on our work.
>
> ### Rebuttal to Reasons To Reject
>
> We would like to first correct a critical misunderstanding.
> > "text files were converted to PDF files in the MULTIEURLEX case"
>
> > "the authors first convert texts into pdf format"
>
> **This is not correct; the original documents used to create MultiEurlex dataset by Chalkidis et al. (2021) [1] comes with PDF files, which we scraped from the EurLex website to build MULTIEURLEX-PDF dataset** (e.g., The example PDF document used in Figure 1 for MULTIEURLEX-DOC is available at https://eur-lex.europa.eu/legal-content/en/TXT/PDF/?uri=CELEX:32006D0213). What we borrowe from the dataset built by [1] are 1) CELEX document IDs (that are used to obtain the corresponding PDF documents from the EurLex website), and 2) the document labels (one or more labels per document due to being multi-label dataset)
>
> > "MultiEURLEX reportedly only has text documents".
>
> Similar to the above point, this is not the case for the original PDF EurLex documents used to create MultiEurlex by Chalkidis et al. (2021) [1]. For example, the document at https://eur-lex.europa.eu/legal-content/EN/TXT/PDF/?uri=CELEX:32006D0213 includes tables and figures which are discarded in the MultiEurlex dataset created by Chalkidis et al. (2021) [1] (which is directly accessible at https://huggingface.co/datasets/multi_eurlex/viewer/en/train?row=0). The structured elements of the documents are not retained in the textual version of the dataset, but this make it very relevant to the problem we are studying.
>
> > "Xu et al (XFUND:A Benchmark Dataset for Multilingual Visually Rich Form Understanding, ACL 2022) reports results of experiments with multi-label classification, zero-shot cross-lingual transfer".
>
> We emphasize here that the tasks explored in Xu et al. 2022 and this paper are different. Xu et al. 2022 report zero-shot cross-lingual transfer results on the semantic entity recognition task and the relation extraction task using the XFUND and FUNSD datasets, but not on the document image classification task. The tasks explored in our work are multi-label and multi-class document image classification (not semantic entity recognition nor relation extraction) and zero-shot cross-lingual transfer on document image classification. Relatedly, we agree with the reviewer that the sentence in Lines 11-14 "Document AI models in previously untested setting such as 1) multi-label classification, and 2) zero-shot cross-lingual transfer setup" is not correctly scoped when read in isolation. We will further clarify the scope of this sentence by scoping only to document image classification task by rephrasing it as "untested for multi-label document image classification and zero-shot cross-lingual transfer on document image classification".
>
> > “D)  [...] the authors claim the datasets need to be hierarchical, multi-label etc”.
>
> We would like to remind that the paper does not claim that datasets need to be hierarchical or multi-label. The goal of the paper is to curate new datasets “to complement the limitations of RVL-CDIP dataset” (Lines 173-175) or specifically, complement the aspect of document image classification task not covered by RVL-CDIP, such as 1) being multi-label in nature (i.e., one or more annotated labels for each document) and 2) covering multiple languages (Lines 150-175.)
>
> We will make all the points listed above clear in the next version of the paper to avoid similar misunderstanding by the future readers of the paper.
>
>
>
> ### Rebuttal to Questions For The Authors
>
> > Ln. 270: What do you mean here? It is used because it was previously used? I was confused. This is not a self-reference is it?
>
> In Line 270, we simply meant to state that we selected InfoXLM as the text-only model following the prior work by Wang et al. 2022.
>
>
> > Ln 400: Where do we observe the result mentioned here? Is it referred to in the tables for few shot learning?
>
> Line 400 from the paper refers to “F1 gap between the two models is significantly smaller in the full-shot setting (90.13 vs. 94.40) than the few-shot setting (26.63 vs. 82.18)“. The full-shot setting is referred from Table 3 (Donut: 90.13, LayoutXLM: 94.40), but as Reviewer BLSd correctly pointed out, the few-shot results in the English WIKI-DOC dataset is not referred in the tables of the paper because the focus of the paper is on non-English experiments, and therefore, we excluded few-shot results on the WIKI-DOC dataset in English.
>
> Here, we show the results we are referring to (i.e., Donut: 26.63, LayoutXLM 82.18) along with other models:
>
>     Models       | Accuracy on WIKI-DOC (with standard deviation across 5 random seeds
>     ------------------------------
>     InfoXLM      | 81.30 (0.85)
>     LiLT         | 81.44 (0.90)
>     LayoutXLM    | 82.18 (1.62)
>     Donut        | 26.63 (3.06)
>
> Thank you for pointing this out and we will add these results in the appendix of the next version of the paper for the sake of completeness.
>
>
>
> > Ln 429: I did not find a table I can refer to for this information.
>
> Line 429 is referring to Table 4 and analyzing the results with respect to language families. Here are the excerpts of Table 4 on Uralic and Slavic language families on mean R-Precision scores (omitting the standard deviations):
>
> Uralic languages:
>
>        | InfoXLM | LayoutXLM
>     ------------------------------
>     hu | 58.84   | 63.87
>     fi | 63.46   | 63.52
>     et | 60.37   | 63.43
>
> Slavic languages:
>
>        | InfoXLM | LayoutXLM
>     ------------------------------
>     pl |  61.12  |  63.26
>     bg |  14.23  |  63.67
>     et |  40.99  |  63.6
>
> From the above excerpts from Table 4, we observe that “LayoutXLM is much better than InfoXLM on Slavic and Uralic languages. ” (Lines 434-435). We will add specific numbers we are referring into in Lines 434-435 for the next version of the paper.
>
>
>
> > Table 8. Can you explain how the correlation is calculated more clearly please?
>
> Thank you for pointing this out. This is explained in Page 8, Footnote 7 and Lines 507-511. To elaborate, the correlation in Table 8 is calculated among the two arrays of numbers. The first array is the accuracy gap between a model trained on English and evaluated on language X and model trained on language X and evaluated on language X. For example, hu→hu scored 58.84 (Table 4), en→hu scored 45.74 (Table 6) and the accuracy gap is 58.84 - 45.74 = 13.10. Focusing now on the syntactic typological feature, the syntactic distance between English and Hungarian is 0.60 (Line 525). We apply similar calculation to the rest of non-Hungarian languages and then calculate the correlation between the array of accuracy gap ([13.10, ...]) and the syntactic distance between English and the target languages ([0.60, ...]) to obtain correlation values in Table 8. The same calculation is done for all typological distances between source and target languages to obtain all correlation values in Table 8. We will add this explanation into the appendix of the next version of the paper.
>
> [1] MultiEURLEX - A multi-lingual and multi-label legal document classification dataset for zero-shot cross-lingual transfer https://aclanthology.org/2021.emnlp-main.559/

---

### Official Review · Reviewer_choU · 2023-08-06

**Typos Grammar Style And Presentation Improvements:** 1. Are these datasets proposed for Do…
**Soundness:** 3

**Excitement:**

3: Ambivalent: It has merits (e.g., it reports state-of-the-art results, the idea is nice), but there are key weaknesses (e.g., it describes incremental work), and it can significantly benefit from another round of revision. However, I won't object to accepting it if my co-reviewers champion it.

**Paper Topic And Main Contributions:**

This paper reported two new multilingual document image datasets for visual document understanding tasks. This paper also evaluates multiple document understanding models on the proposed datasets and analyzes the limitations and challenges proposed by these datasets.

**Reasons To Accept:**

1. The proposed datasets are novel in the document understanding task in terms of the language coverage, and category coverage.
2. The paper performs comprehensive experiments on the datasets and analyzes multiple models' performance on the proposed datasets.
3. The paper reports and analyzes challenges and limitations posed by these datasets which would be inspiring for future research.

**Reasons To Reject:**

1. The writing, especially the result analysis is confusing.
2. The numbers of different language training samples in MULTIEURLEX-DOC dataset are significantly different. However, its influence is not well discussed in the paper.
3. Document images in these datasets are all created by converting pdfs or wiki-pages, which lask coverage of multiple text style, and document style in real applications.

**Reproducibility:**

3: Could reproduce the results with some difficulty. The settings of parameters are underspecified or subjectively determined; the training/evaluation data are not widely available.

**Reviewer Confidence:**

3: Pretty sure, but there's a chance I missed something. Although I have a good feel for this area in general, I did not carefully check the paper's details, e.g., the math, experimental design, or novelty.

---

> ### Author Rebuttal · Authors · 2023-08-28
>
> We thank the reviewer for the comments on our work.
>
> ### Rebuttal to Reasons To Reject
>
> > “The writing, especially the result analysis is confusing”
>
> Our results section (Section 4) is structured such that we first present results on English language for both datasets (Section 4.2). Next we present intralingual results for non-English languages (Section 4.3) and finally we present cross-lingual results (section 4.4). We appreciate more detailed feedback on which specific parts are confusing to further improve the clarity of the paper.
>
>
> > “The numbers of different language training samples [...] significantly different”
>
>
> We did not normalize for the amount of training data to make our results comparable to prior work (i.e Chalkidis et al., 2021). Additionally, our primary goal with intralingual experiments was to compare different models’ performance on the same language or language group (Page 6 right column in Lines 429-435). Furthermore the difference in training data size is only a concern for intralingual experiments and not cross-lingual experiments due to using the model trainined on the single English training set. We agree that the influence of the number of training examples for the intralingual experiments are not mentioned and clarify it in the next version of the paper.
>
>
> > “Document images in these datasets are [...] real applications”
>
>
> The PDFs in both these datasets do have layout variations. The primary focus of this work is not layout understanding but rather understanding the content of the documents where layout also plays an important role. For layout understanding, there is RVL-CDIP dataset where the accuracies of state of the art models have already reached 95-96% thus we choose not to focus on layout understanding since layout understanding is a relatively easy task compared to document content understanding. For example, one of the models (Docformer) achieves ~96% accuracy on RVL-CDIP however it only achieves mRP of 63.46 on MULTIEURLEX-DOC (Table 3).
>
>
> ### Rebuttal to Typos Grammar Style And Presentation Improvements
> > “Are these datasets proposed for Document AI, or document classification task?”
>
>
> We will improve the naming convention in final version of the paper to make it consistent. As mentioned earlier, our focus is more on understanding content of the semi-structured documents rather than the high level layout of these document since there is already an existing dataset for layout understanding (RVL-CDIP).
>
>
> > “Line 439 - 440. This sentence is inaccurate and confusing”
>
>
> These lines talk about results on WIKI-DOC specifically on Germanic languages like German (de) and romance languages like Spanish (es), French (fr), Italian (it) and Portuguese (pt) based on results in Table 5 so it is not inaccurate.
>
>
> > “The expressions in Sec 4.4.1 are confusing”
>
> We kindly request Reviewer choU for adding further details on which expressions in Section 4.4.1 are confusing so that we can rephrase and clarify for the future readers of the paper.

---

### Meta-Review · Area_Chair_fpNa · 2023-09-20

**Recommendation:** 4

**Metareview:**

Pros:
- This paper introduces two new multilingual document image classification datasets: MULTIEURLEX-DOC and WIKI-DOC
- "The paper reports and analyzes challenges and limitations posed by these datasets which would be inspiring for future research."

Cons:
- The details on the proposed dataset and experimental setup are highlighted to be confusing and can be made clearer.

---

### Decision · Program_Chairs · 2023-10-07

**Decision:**

Accept-Findings

**Comment:**

Pros:
- This paper introduces two new multilingual document image classification datasets: MULTIEURLEX-DOC and WIKI-DOC
- "The paper reports and analyzes challenges and limitations posed by these datasets which would be inspiring for future research."

Cons:
- The details on the proposed dataset and experimental setup are highlighted to be confusing and can be made clearer.